# Relationship between lymphedema after breast cancer treatment and biophysical characteristics of the affected tissue

Carla S. Perez[1,2☯*], Carolina Mestriner[1☯], Leticia T. N. Ribeiro[3], Felipe W. Grillo[3], Tenysson W. Lemos[2], Antônio A. Carneiro[3], Rinaldo Roberto de Jesus Guirro[1,2], Elaine C. O. Guirro[1,2]*

1 Programa de Pós-Graduação em Reabilitação e Desempenho Funcional, Universidade de São Paulo, Ribeirão Preto, São Paulo, Brasil, 2 Departamento de Ciências das Saúde, Faculdade de Medicina de Ribeirão Preto, Universidade de são Paulo, Ribeirão Preto, São Paulo, Brasil, 3 Departamento de Física, Faculdade de Filosofia, ciências e Letras de Ribeirão Preto, Universidade de São Paulo, Ribeirão Preto, São Paulo, Brasil

☯ These authors contributed equally to this work.
* carlaperez@alumni.usp.br, carlaperez86@gmail.com (CSP); ecguirro@fmrp.usp.br (ECOG)

**Data Availability Statement:** All relevant data are available via https://doi.org/10.5281/zenodo.4549683 (DOI: 10.5281/zenodo.4549683).

## Abstract

The treatment of breast cancer is often complicated by lymphedema of the upper limbs. Standard lymphedema evaluation methodologies are not able to measure tissue fibrosis. The ultrasound aspects related to tissue microstructures of lymphedema are neglected in clinical evaluations. The objective of this study was to identify and measure the degree of impairment, topography, and biophysical alterations of subcutaneous lymphedema tissue secondary to the treatment of breast cancer by ultrasonography. Forty-two women at a mean age of 58 (±9.7) years, with unilateral lymphedema due to breast cancer treatment, were evaluated. The upper limbs were divided into affected (affected by lymphedema) and control (contralateral limb). Each limb was subdivided into seven areas, defined by perimetry, evaluated in pairs. The biophysical characteristics thickness, entropy, and echogenicity were evaluated by ultrasonography. The results showed a significant difference in the echogenicity and thickness variables between the affected and unaffected upper limb, in all the extent of the upper limb, while entropy showed no significant difference. The findings indicate that the data presented were consistent both in identifying and measuring the degree of impairment and biophysical changes in the subcutaneous tissue of lymphedema secondary to the treatment of breast cancer.

## Introduction

Lymphedema secondary to the treatment of breast cancer is a chronic and recurrent condition involving the lymphatic and blood systems [1,2]. The dysfunctional lymphatic system becomes less capable of performing the complete resorption of large protein molecules, and these remain in the interstitial space. The consequent tissue fibrosis and the increasing accumulation

**Funding:** This research did not receive any specific grant from funding agencies in the public, commercial, or not-for-profit sectors.

**Competing interests:** The authors have declared that no competing interests exist.

of fluid and proteins in this space can trigger neurological alterations such as pain or paresthesia, distortion in the shape of the limb, and increased risk of related complications [3,4]. Chronic lymphedema causes physical deficiencies and psychological stress, which worsens with the progression of the dysfunction, to reduce the discomfort of the patient and improve the quality of life, an accurate diagnosis of lymphedema is essential for prognosis and treatment planning [5].

The evaluation of limb volume is considered the gold standard for measuring irregular edema and is considered the main routine measure for the diagnosis and evaluation of the evolution of lymphedema treatments [6,7]. However, these measures only characterize the external shape of the related upper limb, not having the sensitivity necessary to identify the intrinsic alterations of the affected tissue. The exclusively clinical diagnosis of lymphedema does not indicate consistent thresholds for the difference between the upper limbs, or between different evaluators. Subjective measures such as palpation and patient perception are not consistent in the evaluation of lymphedema [8].

Other medical imaging modalities are applied to diagnose lymphedema [9,10]. Among these, lymphoscintigraphy is the main imaging modality to evaluate circulatory dysfunctions of the lymphatic system, considered the standard for decades [11]. Nonetheless, it does not identify biophysical alterations such as stiffness, topography, and size of the limb with lymphedema, one of the main causes of discomfort and impairment of limb function in patients [12].

Chronic lymphedema induces complications such as inflammation, fat tissue hypertrophy, fibrosis, and recurrent infections [13]. These variables can be taken into account in the analysis of the ultrasound image of the affected tissue [8,14]. The topography, texture, and stiffness of the tissue that may directly influence the responses resulting from different therapeutic interventions have not been established yet. The control of lymphedema is fundamental, as it is considered a definitive condition [4], to prevent related comorbidities, minimizing complications with adequate treatment, as well as helping to monitor the progression of the dysfunction, assisting in the prognosis, and in the evaluation of effects resulting from different therapeutic interventions.

The objective of this study was to identify and measure the degree of impairment, topography, and biophysical alterations of subcutaneous lymphedema tissue secondary to the treatment of breast cancer by ultrasonography. For that, thickness, entropy, and echogenicity in seven points of the upper limb were examined and measured, with a subsequent comparison between the affected and unaffected upper limbs.

## Materials and methods

### Participants

The study was designed a type of cohort design of women enrolled between December/2017 and December/2018. The patients were recruited at REMA (Center for Teaching, Research, and Assistance in the Rehabilitation of Mastectomized Patients) at the University of São Paulo at Ribeirão Preto, College of Nursing (EERP-USP). Interested individuals completed a personal screening of the disease and treatment. 125 women with lymphedema from breast cancer treatment were invited and elected to participate in this study, 83 of them were excluded by the established criteria and 42 patients were allocated in this study. All participants provided institutionally approved, written, informed consent under a study protocol approved by the Ethics and Research Committee of the Medical School of Ribeirão Preto of the University of São Paulo (FMRP/USP), CAAE Process 65981216.4.0000.5440.

The sample was calculated based on the study by Suehiro *et al.* (2014) [15], with a statistical power of 80% and an alpha error of 0.05. The program used was StateMate 2 (GraphPad

Software® v 2.0), resulting in n = 35, and to compensate for possible losses, a sample of 42 was considered.

The outcome of the measurement of lymphedema was obtained through indirect measurement of volume, determined by the upper limb's circumference. The limb volume was calculated from the circumference measurements, treating each segment of the limb as a pair of circumferences, formed by the measurement points of the circumference of the seven points of the arm and forearm, called truncated cones. Lymphedema was considered when there was a difference greater than 2 cm in the perimetry of two or more predetermined points on the affected limb compared to the contralateral limb [16].

Inclusion criteria were women over 21 years old who underwent treatment for unilateral breast cancer, had unilateral lymphedema from treatment for breast cancer, according to criteria of Sander *et al.*, (2001). Women with skin conditions, adjuvant treatment in progress, diagnosis of metastasis or unexplained edema in the arm not involved were excluded. None of the participants was under intensive treatment for lymphedema.

## Lymphedema measurements

**Assessment of member volume.** Limb volume was evaluated by indirect measurement, resulting from the evaluation of the perimetry of both upper limbs and the sum of the approximate volume of six truncated cones, formed of the measurement of the circumferences of the seven points on the arm and forearm. The sum of these six parts provided the total volume of the limb. The method of measuring the indirect volume has good levels of intra- and inter-examiner reliability, with values of intraclass correlation coefficient (ICC) of 0.99 [16].

**Evaluation of biophysical characteristics by ultrasound.** Ultrasound images (US) were obtained using Ultrasonics ultrasound equipment (Sonix RP, Richmond, BC, Canada), with a linear transducer of 9–15 MHz, gain 70%, depth 5.0 cm. The B-mode image and the RF (radio frequency) signal were collected simultaneously. A phantom (biological tissue-mimicking material), produced from a 2.5 cm thick styrene—ethylene—butylene—styrene copolymer (SEBS), was used between the transducer and the skin, both surfaces coupled with ultrasound gel, to standardize the collections.

The signal was collected in the anterior region of the affected upper limb and of the unaffected limb by lymphedema in the seven points already pre-defined for perimetry, we considered a line that was made to standardize these points starting from the acromion and passing through the midpoint between the epicondyles and the styloid process of the radius and ulna. Other studies that quantify the characteristics of the subcutaneous tissue use the anterior region of the forearm [17,18]. This is a relatively level surface where there is the convenience of positioning the US transducer (Fig 1).

In each US image, the region of interest (ROI) was defined as the distance between two echogenic lines, as described in previous studies defining the subcutaneous tissue, the thickness of this region, and the measurements of echogenicity and entropy [19–21].

The ultrasound signal from the affected and unaffected upper limb, 30 frames were obtained for each point, for each of the seven predefined points. These images were saved in both B and RF (Radiofrequency) modes. For this analysis, the B mode (.b8) file was used. For processing the ultrasound images, computational routines in the Matlab environment (The-MathWorks, Inc., Natick, MA) were used to obtain three images for each point and transform them into another extension for analysis (.png). The images in (.png) were analyzed using the Image J software. The Straight function present in the software to measure distance was used to quantify the thickness of the tissue affected by lymphedema in pixels. For entropy and echogenicity analysis, ROI was initially defined using the *Polygon selection* tool. In this ROI,

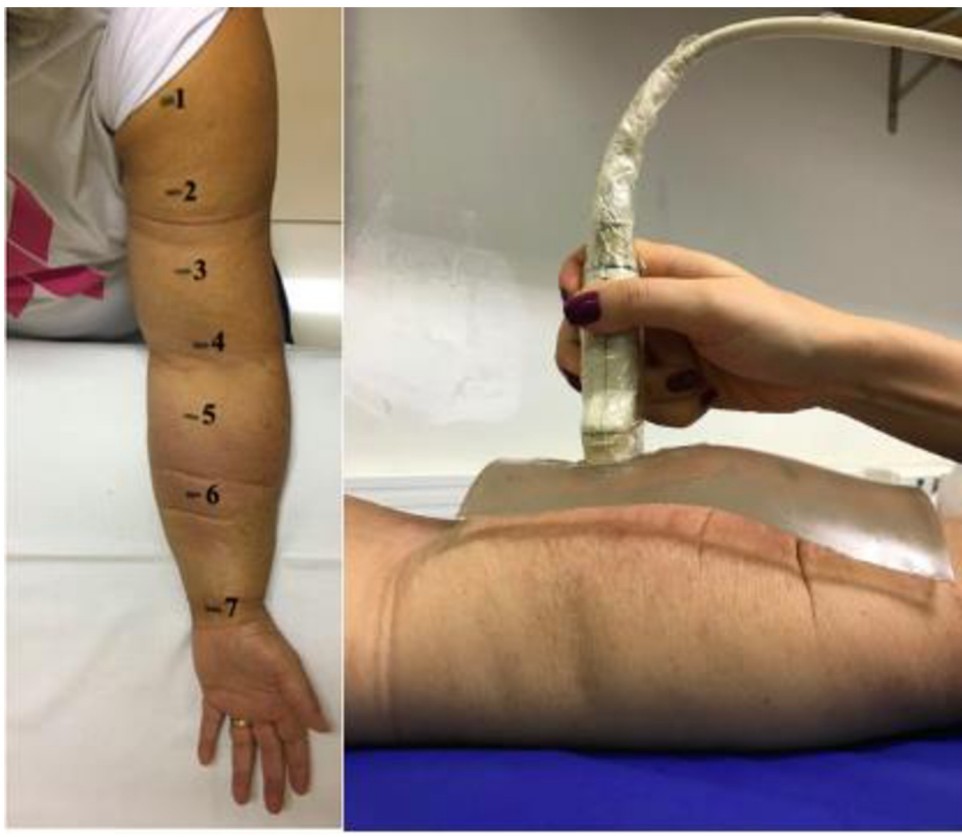

**Fig 1. a) Illustrative image of the points used for elastography; b) Placement of the US pad and transducer.**
**Source**: Personal records.

standard measurements of entropy and echogenicity were implemented with a new plugin that was added and used in the software (Fig 2).

## Statistical analysis

The data were submitted to exploratory analysis, which had the primary objective of synthesizing a series of values of the same nature, allowing a global view of the variation of these values, organizing and describing the data in tables with descriptive measures.

After tabulating the variables, the Shapiro-Wilk normality test was applied to analyze the distribution. The echogenicity and entropy variables presented non-parametric distribution; when Wilcoxon was applied. The thickness variable presented normal distribution, so the related T-test was applied. In all calculations, a critical level of 5% ($p < 0.05$) was fixed, and data processing was performed by SPSS software, version 17.0.

## Results

The general characteristics and the characteristics relative to the surgical treatment of patients submitted to the evaluation of lymphedema by ultrasound are present in Table 1.

The results regarding echogenicity (capacity of the tissue to reflect an echo of the ultrasound wave, indicative of the density of the subcutaneous tissue) of the upper limb affected and unaffected by lymphedema are present in Table 2. The affected upper limb showed greater

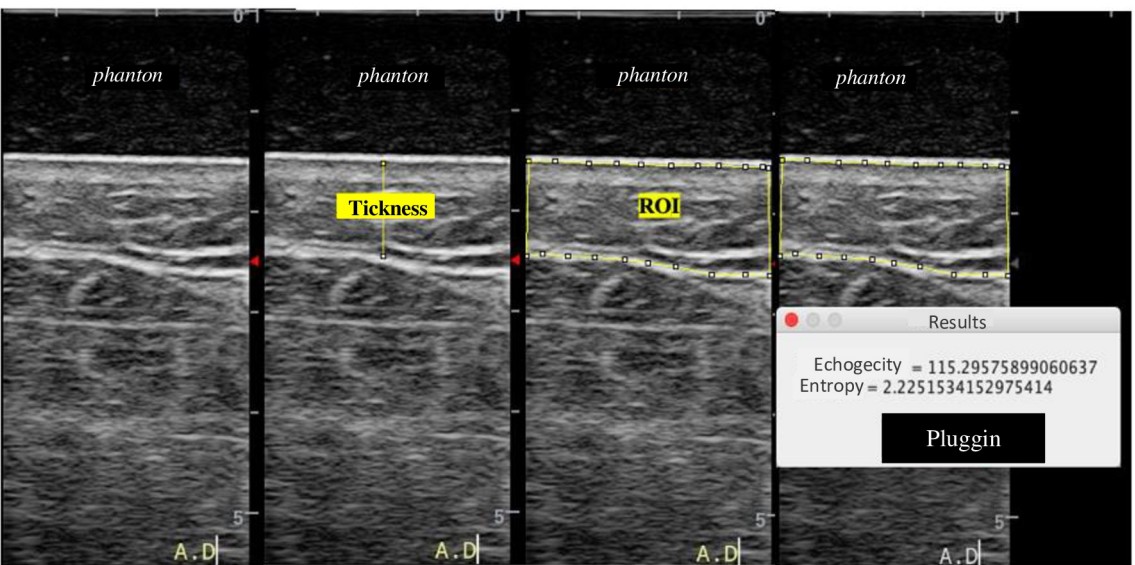

**Fig 2. Image of thickness, echogenicity and entropy evaluation in Image J. Source**: Personal records.

echogenicity at all points, except for point 7 (wrist), probably due to the presence of tendons in the region.

The results regarding entropy (a measurement of subcutaneous tissue disorder) of the upper limb affected and unaffected by lymphedema are present in Table 3. This variable did not vary significantly among limbs affected or not.

The data regarding the thickness of the subcutaneous tissue of the upper limb affected by lymphedema was significantly higher at all points except at point 4 (elbow), compared to the unaffected upper limb (Table 4).

## Discussion

Fibrosis is a histological characteristic of lymphedema; however, its measurement in the patient sometimes is still performed through clinical measures, such as palpation [22]. This

**Table 1. General characteristics of patients with lymphedema.**

| Characteristics | | |
|---|---|---|
| Age (years) | 58 (9.7) | |
| BMI (Kg/m$^2$) | 31.86 (6.05) | |
| Type of surgery | Mastectomy | 24 |
| | Quadrantectomy | 10 |
| | Nodulectomy | 8 |
| Side Surgery | Right | 23 |
| | Left | 19 |
| Axillary Lymphadenectomy | Yes | 42 |
| | No | 0 |

*Values presented as mean (standard deviation). BMI = Body mass index.

**Table 2. Values referring to echogenicity in the affected upper limb (MSA) and unaffected upper limb (MSNA) at the seven predetermined points.**

| Points | MSA | MSNA | p |
|---|---|---|---|
| 1 | 51.94 (41.51, 50.13) | 44.91 (37.88, 63.87) | <0.001 |
| 2 | 56.76 (47.72, 78.74) | 54.76 (45.68, 70.11) | 0.001 |
| 3 | 61.55 (51.25, 81.28.) | 57.07 (48.60, 73.35) | <0.001 |
| 4 | 55.34 (46.94, 93.82) | 55.54 (42.08, 75.42) | 0.004 |
| 5 | 52.28 (42.66, 83.55) | 51.89 (42.08, 75.06) | 0.040 |
| 6 | 53.49 (42.91, 89.31) | 47.28 (39.05, 68.01) | 0.001 |
| 7 | 62.32 (53.09, 88.69) | 64.00 (53.70, 92.70) | 0.161 |

*Values presented in median (first quartile, third quartile). MSA: Affected upper limb by lymphedema. MSNA: Unaffected upper limb by lymphedema.

study aimed to measure fibrosis of the subcutaneous tissue in lymphedema secondary to the treatment of breast cancer by ultrasonography.

The results showed that both thickness and echogenicity are measured through the differentiation between the upper limb affected by lymphedema and the unaffected upper limb, showing all the degree of impairment of the affected tissue either by the increase in size, in case of thickness, or fibrosis developed, quantified by employing the echogenicity variable. These findings demonstrate how the use of ultrasound images can be a differential in the diagnosis and treatment of lymphedema. Once this is a definitive condition with negative repercussions on physical, psychological, and emotional well-being, lymphedema must be controlled to prevent comorbidities [4]. Ultrasonography is considered a promising method [8] and our results confirm that this tool is effective in the evaluation and characterization of biophysical alterations inherent to lymphedema.

Understanding the degree of structural involvement of the affected tissue can help to monitor the progression of the dysfunction, assisting in the prognosis, as well as assessing the effects of different therapeutic interventions. In the present study, there was a concern about analyzing the lymphedema distribution topography; the findings showed that the regions closest to the elbow presented greater differences, unlike from the wrist in terms of the amount of fibrosis.

**Table 3. Values referring to entropy in the affected upper limb (MSA) and unaffected upper limb (MSNA) at the seven predetermined points.**

| Points | MSA | MSNA | p |
|---|---|---|---|
| 1 | 1.87 (1.84, 1.99) | 1.89 (1.83, 2.07) | 0.061 |
| 2 | 1.89 (1.85, 2.08) | 1.89 (1.85, 2.04) | 0.472 |
| 3 | 1.89 (1.86, 2.07) | 1.88 (1.85, 2.08) | 0.091 |
| 4 | 1.89 (1.82, 1.99) | 1.87 (1.85, 2.08) | 0.738 |
| 5 | 1.88 (1.85, 2.03) | 1.89 (1.85, 2.07) | 0.965 |
| 6 | 1.88 (1.85, 2.03) | 1.88 (1.83, 2.02) | 0.574 |
| 7 | 1.87 (1.83, 2.01) | 1.84 (1.82, 2.05) | 0.213 |

*Values presented in median (first quartile, third quartile). MSA: Affected upper limb by lymphedema. MSNA: Unaffected upper limb by lymphedema.

**Table 4. Pixel thickness values of the subcutaneous tissue in the affected upper limb (MSA) and unaffected upper limb (MSNA) at the seven predetermined points.**

| Points | MSA | MSNA | p |
|--------|-----|------|---|
| 1 | 99.36 (31.62) | 84.59 (36.32) | <0.001 |
| 2 | 94.34 (32.94) | 78.81 (38.37) | 0.009 |
| 3 | 83.16 (34.39) | 62.99 (34.98) | <0.001 |
| 4 | 102.50 (212.37) | 50.60 (26.32) | 0.110 |
| 5 | 87.73 (32.46) | 56.61 (25.47) | <0.001 |
| 6 | 102.87 (35.80) | 74.90 (40.44) | <0.001 |
| 7 | 69.24 (34.22) | 48.56 (34.36) | <0.001 |

*Values presented as mean (standard deviation). MSA: Affected upper limb by lymphedema. MSNA: Unaffected upper limb by lymphedema.

Due to fibrosis and the blocking of lymphatic channels in the regions near the elbow of an upper limb affected by lymphedema, there is an important impairment of lymphatic circulation. The lymphatic pathway of this limb presents significant differences such as obliteration of superficial lymphatic vessels, dermal reflux, uncommon communication between the superficial and deep lymphatics, and interval lymph nodes. These alterations seem to facilitate lymphatic drainage after sporadic blockage of the lymph, especially proximal in the limb [23].

The circulatory function is influenced by the drainage of superficial and deep lymphatic vessels and is often interrupted in some regions of lymphedema, triggering superficial collateralization with retrograde flow to the lymphatic vessels of the skin. Lymphatic fluid stasis is associated with the accumulation of interstitial fluid in the subcutaneous tissue and skin and also with an accumulation of proteins and glycosaminoglycans, the retained interstitial fluid subsequently stimulates the production of collagen, which leads to skin thickness and subcutaneous fibrosis of soft tissues [24].

Subjective measures such as palpation and patient perception are not consistent for assessing lymphedema fibrosis. Furthermore, the diagnosis exclusively based on volume does not point to consistent thresholds for the difference between the upper limbs, or for the evaluation of professionals [8]. In *phantom* testing in a reduced number of patients using 2D ultrasound, there were encouraging initial results; deformation values of the arms affected by lymphedema were much higher than those of normal arms [25]. In the present study, it is possible to observe a difference in the thickness of the subcutaneous tissue by ultrasound. Although volume and palpation are routinely measured for diagnosis and evaluation of treatment evolution, they are very inconsistent, supporting the argument that they should not be the only objective measurement of lymphedema.

Both entropy, a measure of randomness within the ultrasound image reflecting the degree of organization or disorganization of tissue, and echogenicity, a measure of the capacity of the tissue reflecting an echo of the ultrasound wave, serve as an indicator of the density of the tissue and may be able to discriminate the affected and unaffected side of lymphedema [8,14]. Ashikaga *et al*. (2005) showed a preference for the measurement of entropy to characterize image texture, which was seen by increasing the values of entropy on the affected side. However, Johson *et al*. (2016) observed a lower entropy. The difference between these studies is the different types of surgery evaluated.

Entropy measurements were less variable in this study, so it was not significant between members. The previously mentioned studies only evaluated a predetermined *ROI* (region of interest) of a few millimeters. This study is a pioneer in evaluating the entire subcutaneous

tissue area of the assessed points. This choice may have been a determinant of why the difference between limbs was not significant in the entropy variable, as this is a randomization measure of the ultrasound image, which reflects how much the tissue is organized or not [26]. Therefore, larger regions could have more variations, once in the subcutaneous tissue there are also other structures such as fat, and not only fibrosis. It is important to note that most of the population studied are overweight or obese, which is common in surviving breast cancer patients [12,27].

Echogenicity is a measure of the ability of the tissue to reflect an echo of the ultrasound wave, higher echogenicity indicates that the tissue has a higher density suggestive of higher fibrosis. Studies [28,29] have shown that upper limbs with lymphedema resulting from the treatment of breast cancer increased or altered the echogenicity in subcutaneous tissue. However, the authors did not quantify the variable. The measurement of echogenicity through subcutaneous ultrasonography can improve the accuracy of diagnosis and can be used to monitor the progress and severity of lymphedema in the forearm [30]. The present study quantified and demonstrated that the analysis of echogenicity can be an important tool in the evaluation of the degree of fibrosis of the affected upper limb.

Previous studies [14,26] have only evaluated two measurement points, one in the arm and other in the forearm, or even an additional measurement close to the elbow [8]. To achieve the proposed objectives, this study included seven measurement points, necessary to evaluate irregular edemas such as lymphedema. Differences were observed in our results such as a significant increase in the thickness of the subcutaneous tissue measured by ultrasonography in the upper limb, different in distinct areas affected by lymphedema.

Previous studies [8,14] have used 4 mm of depth for the analysis of their images, but because this ROI is too small, it only registers skin and the most superficial region of the subcutaneous tissue. This limitation was observed by Johson *et al*. (2016), who considers that important information regarding the subcutaneous tissue may have been lost. In the present study, the 5.0 cm depth was used, thus, excluding the possibility of a loss of information regarding the subcutaneous tissue. The *phantom* thickness of 2.5 cm is another reason for the choice of using a depth of 5.0 cm.

One limitation of the tissue was the use of a transducer smaller than 20 MHz, which presents a lower spatial resolution and consequent visualization of structures close to it. The resolution for this limitation was to make a *phantom* used as a spacer between the transducer and the skin to minimize potential resolution problems that would affect the accuracy of the images since it is a transducer smaller than 20 MHz.

The understanding of the involvement of the tissue affected by lymphedema, in a thorough way, can help to monitor the progression of the dysfunction, assisting in the prognosis, as well as in the evaluation of effects resulting from different therapeutic interventions. The diagnostic ultrasound technique can help understand how the treatments act on the intrinsic tissue affected by lymphedema, which is fundamental for both clinical performance and research. Therefore, the analysis of the ultrasound image seems to be a promising tool for both the evaluation and monitoring of different interventions.

## Conclusion

The findings indicated that the data presented were consistent in identifying the biophysical alterations of subcutaneous lymphedema tissue secondary to the treatment of breast cancer. The thickness of the subcutaneous tissue was increased in all measurements except for the elbow, which is not such a complacent region. However, entropy in these parameters still

showed no difference between the affected and unaffected upper limbs. Echogenicity, quanti-
fied to measure fibrosis was increased in all upper limbs except for the wrist.

## Author Contributions

**Conceptualization:** Carla S. Perez, Leticia T. N. Ribeiro, Felipe W. Grillo, Tenysson W.
Lemos, Antônio A. Carneiro, Elaine C. O. Guirro.

**Data curation:** Carla S. Perez, Felipe W. Grillo, Antônio A. Carneiro, Rinaldo Roberto de
Jesus Guirro, Elaine C. O. Guirro.

**Formal analysis:** Carla S. Perez, Leticia T. N. Ribeiro, Antônio A. Carneiro, Rinaldo Roberto
de Jesus Guirro, Elaine C. O. Guirro.

**Investigation:** Carla S. Perez, Carolina Mestriner, Leticia T. N. Ribeiro, Tenysson W. Lemos,
Antônio A. Carneiro, Rinaldo Roberto de Jesus Guirro, Elaine C. O. Guirro.

**Methodology:** Carla S. Perez, Carolina Mestriner, Leticia T. N. Ribeiro, Felipe W. Grillo,
Tenysson W. Lemos, Antônio A. Carneiro, Rinaldo Roberto de Jesus Guirro, Elaine C. O.
Guirro.

**Project administration:** Carla S. Perez, Carolina Mestriner, Tenysson W. Lemos, Antônio A.
Carneiro, Elaine C. O. Guirro.

**Resources:** Carla S. Perez, Antônio A. Carneiro, Elaine C. O. Guirro.

**Software:** Carla S. Perez, Leticia T. N. Ribeiro, Felipe W. Grillo, Tenysson W. Lemos, Antônio
A. Carneiro, Rinaldo Roberto de Jesus Guirro, Elaine C. O. Guirro.

**Supervision:** Carla S. Perez, Antônio A. Carneiro, Elaine C. O. Guirro.

**Validation:** Carla S. Perez, Felipe W. Grillo, Antônio A. Carneiro, Elaine C. O. Guirro.

**Visualization:** Carla S. Perez, Antônio A. Carneiro, Elaine C. O. Guirro.

**Writing – original draft:** Carla S. Perez, Carolina Mestriner, Leticia T. N. Ribeiro, Felipe W.
Grillo, Tenysson W. Lemos, Antônio A. Carneiro, Rinaldo Roberto de Jesus Guirro, Elaine
C. O. Guirro.

**Writing – review & editing:** Carla S. Perez, Carolina Mestriner, Leticia T. N. Ribeiro, Felipe
W. Grillo, Tenysson W. Lemos, Antônio A. Carneiro, Rinaldo Roberto de Jesus Guirro,
Elaine C. O. Guirro.

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
