## [Decision Letter · Decision Letter 0]

18 Jan 2021

PONE-D-20-33153

RELATIONSHIP BETWEEN LYMPHEDEMA AFTER BREAST CANCER TREATMENT AND BIOPHYSICAL CHARACTERISTICS OF THE AFFECTED TISSUE

PLOS ONE

Dear Dr. Perez,

Thank you for submitting your manuscript to PLOS ONE. After careful consideration, we feel that it has merit but does not fully meet PLOS ONE’s publication criteria as it currently stands. Therefore, we invite you to submit a revised version of the manuscript that addresses the points raised during the review process.

Two experts raised several concerns. The study design and results does not support your conclusions.  The authors should revise extensively in data analysis and discussion to draw the conclusion. 

We look forward to receiving your revised manuscript.

Kind regards,

Tatsuo Shimosawa, M.D., Ph.D.

Academic Editor

PLOS ONE

Journal Requirements:

2. Please provide additional details regarding participant consent. In the ethics statement in the Methods and online submission information, please ensure that you have specified (1) whether consent was informed and (2) what type you obtained (for instance, written or verbal, and if verbal, how it was documented and witnessed). If your study included minors, state whether you obtained consent from parents or guardians. If the need for consent was waived, please ensure that you have discussed whether all data were fully anonymized before you accessed them and/or whether the IRB or ethics committee waived the requirement for informed consent.

3. In your Methods section, please provide additional information about the participant recruitment method and the demographic details of your participants. Please ensure you have provided sufficient details to replicate the analyses such as: a) a description of how participants were recruited, and b) descriptions of where participants were recruited and where the research took place.

4.Thank you for stating the following in the Acknowledgments Section of your manuscript:

"This study was financed in part by the Coordenação de Aperfeiçoamento de Pessoal

de Nível Superior - Brasil (CAPES) - Finance Code 88881.190398/2018-01."

Reviewers' comments:

Reviewer's Responses to Questions

**Comments to the Author**

1. Is the manuscript technically sound, and do the data support the conclusions?

Reviewer #1: Yes

Reviewer #2: Partly

2. Has the statistical analysis been performed appropriately and rigorously? 

Reviewer #1: Yes

Reviewer #2: Yes

3. Have the authors made all data underlying the findings in their manuscript fully available?

Reviewer #1: Yes

Reviewer #2: Yes

4. Is the manuscript presented in an intelligible fashion and written in standard English?

Reviewer #1: Yes

Reviewer #2: Yes

5. Review Comments to the Author

Reviewer #1: Perez et al. presented the evaluation of lymphedema after breast cancer by using ultrasonography. They focused on tissue thickness, echogenicity, and entropy. These components seem to be a kind of worn-out technique. Still, it is of note that they mention fibrosis caused by lymphedema. Once fibrosis has developed, it is quite difficult to treat it. To choose treatment options, their technique may be useful in determining whether fibrosis is already well established or not. However, I have several major concerns about this report the authors should meet.

Major concern 1

The authors evaluated values of ultrasonography on lymphedema, just compared with unaffected side of upper limbs. It is not so difficult to decide whether there is lymphedema or not via subjective measures. It is much more important to evaluate how much lymphedema progresses. Therefore, they need to illustrate the association between values of ultrasonography and the progression of lymphedema. The author should analyze the differences between patients. They need to illustrate whether the index proposed in the manuscript can reflect the severity of the lymphedema.

Major concern 2

They stated that clinical problem in lymphedema after breast cancer lies in lack of diagnostic tools, but there are actually much more serious problems, including the difficulty in controlling lymphedema and stopping the fibrosis, and lack of treatment options. They need to point out how their technique would contribute to the future treatment of lymphedema.

Minor point

Tissue Elastgraphy is also attracting more and more attention as a non-invasive measurement for fibrosis. Does the author’s method correlate to these other methods?

Reviewer #2: This study showed the echogenicity and the thickness of subcutaneous tissue of the upper limb on ultrasonography (US) related to lymphedema after breast cancer treatment.

But there are some problems on this manuscript.

1) What is the definition of lymphedema? Please describe in the participant section.

2) In the last paragraph of the Discussion, the authors say that the US seems to be a promising tool for the monitoring of interventions, but this study did not assess the changes of US findings after treatment for lymphedema and the prognosis.

3) This study did not include the degree of impairment, therefore the first sentence of the conclusion seems inadequate.

Minor points

1) Table 1 law means right?

2) Table 2 -4, abbreviation of MSA and MSNA should be spelled out in the foot note.

6. PLOS authors have the option to publish the peer review history of their article (what does this mean?). If published, this will include your full peer review and any attached files.

Reviewer #1: No

Reviewer #2: No

---

## [Author Response · Author response to Decision Letter 0]

18 Feb 2021

Thank you for your valuable inputs towards our study. Therefore, the corrections respond to the recommendations of all reviewers. They emphasized the relevant data and enhanced comprehension of the manuscript.

Journal Requirements:

Answer: The manuscript meets PLOS ONE's style requirements, including those for file nomenclature. 

2. Please provide additional details regarding participant consent. In the ethics statement in the Methods and online submission information, please ensure that you have specified (1) whether consent was informed and (2) what type you obtained (for instance, written or verbal, and if verbal, how it was documented and witnessed). If your study included minors, state whether you obtained consent from parents or guardians. If the need for consent was waived, please ensure that you have discussed whether all data were fully anonymized before you accessed them and/or whether the IRB or ethics committee waived the requirement for informed consent.

Answer: All participants provided institutionally approved, written, informed consent under a study protocol approved by the Ethics and Research Committee of the Medical School of Ribeirão Preto of the University of São Paulo (FMRP/USP), CAAE Process 65981216.4.0000.5440. This information was inserted in the manuscript and is highlighted in the document 'Revised Manuscript with Track Changes'. 

3. In your Methods section, please provide additional information about the participant recruitment method and the demographic details of your participants. Please ensure you have provided sufficient details to replicate the analyses such as: a) a description of how participants were recruited, and b) descriptions of where participants were recruited and where the research took place.

Answer: The patients were recruited at REMA (Center for Teaching, Research, and Assistance in the Rehabilitation of Mastectomized Patients) at the University of São Paulo at Ribeirão Preto, College of Nursing (EERP-USP). Interested individuals completed a personal screening of the disease and treatment. This information was inserted in the manuscript and is highlighted in the document 'Revised Manuscript with Track Changes'. 

4.Thank you for stating the following in the Acknowledgments Section of your manuscript:

"This study was financed in part by the Coordenação de Aperfeiçoamento de Pessoal de Nível Superior – Brasil (CAPES) – Finance Code 001"."

Answer: The information was duly removed from the acknowledgments: 

Reviewer's Responses to Questions

Comments to the Author

Is the manuscript technically sound, and do the data support the conclusions?

Reviewer #1: Yes

Reviewer #2: Partly

3. Has the statistical analysis been performed appropriately and rigorously? 

Reviewer #1: Yes

Reviewer #2: Yes

4. Have the authors made all data underlying the findings in their manuscript fully available?

Reviewer #1: Yes

Reviewer #2: Yes

5. Is the manuscript presented in an intelligible fashion and written in standard English?

Reviewer #1: Yes

Reviewer #2: Yes

Review Comments to the Author

Reviewer #1: Perez et al. presented the evaluation of lymphedema after breast cancer by using ultrasonography. They focused on tissue thickness, echogenicity, and entropy. These components seem to be a kind of worn-out technique. Still, it is of note that they mention fibrosis caused by lymphedema. Once fibrosis has developed, it is quite difficult to treat it. To choose treatment options, their technique may be useful in determining whether fibrosis is already well established or not. However, I have several major concerns about this report the authors should meet.

Major concern 

1. The authors evaluated values of ultrasonography on lymphedema, just compared with unaffected side of upper limbs. It is not so difficult to decide whether there is lymphedema or not via subjective measures. It is much more important to evaluate how much lymphedema progresses. Therefore, they need to illustrate the association between values of ultrasonography and the progression of lymphedema. The author should analyze the differences between patients. They need to illustrate whether the index proposed in the manuscript can reflect the severity of the lymphedema.

Answer: The data analyzed in diagnostic ultrasonography aimed to characterize thickness, echogenicity, and entropy measurements and their distribution in the upper limb affected and not affected by lymphedema. Subjective measures still seem insufficient to identify lymphedema since most measures are clinical and dependent evaluators (Johnson et al., 2016).

Regarding the measures presented and the disease progression, the greater the thickness, the greater the limb's volume, echogenicity may be related to the amount of collagen tissue, suggesting fibrosis, and the greater the entropy, the more significant tissue disorganization. To establish this relationship, the measurement of these measures is done first between the affected tissue and a phantom of known measures. Only after establishing this relationship is a comparison between the limbs. 

The choice of a longitudinal study appears to be interesting, but it does not include this study's objectives. Lymphedema is a chronic disease, where the time of onset is still unknown, many patients take months to years to seek health care. Also, several factors can alter the volume and tissue fibrosis of lymphedema during the course of the disease, such as injuries, physical inactivity, and weight change, among others. Therefore, the data cannot infer the disease's progression since it is multifactorial.

Johnson KC, DeSarno M, Ashikaga T, Henry SM. Ultrasound and clinical measures for lymphedema. Lymphat Res Biol. 2016;14(1):8-17.

Major concern 

2. They stated that clinical problem in lymphedema after breast cancer lies in lack of diagnostic tools, but there are actually much more serious problems, including the difficulty in controlling lymphedema and stopping the fibrosis, and lack of treatment options. They need to point out how their technique would contribute to the future treatment of lymphedema.

Answer: The treatments for lymphedema concentrate only on controlling the upper limb's volume, with little focus on tissue alteration. The diagnostic ultrasound technique can help understand how the treatments act on the intrinsic tissue affected by lymphedema, which is fundamental for both clinical performance and research. This information was inserted in the manuscript and is highlighted in the document 'Revised Manuscript with Track Changes'. 

Minor point

Tissue Elastgraphy is also attracting more and more attention as a non-invasive measurement for fibrosis. Does the author's method correlate to these other methods?

Answer: The data may have a complementary relationship since elastography would measure the "hardness", replacing the palpation. 

Reviewer #2: This study showed the echogenicity and the thickness of subcutaneous tissue of the upper limb on ultrasonography (US) related to lymphedema after breast cancer treatment.

But there are some problems on this manuscript

1) What is the definition of lymphedema? Please describe in the participant section.

Answer: The outcome of the measurement of lymphedema was obtained through indirect measurement of volume, determined by the upper limb's circumference. The limb volume was calculated from the circumference measurements, treating each segment of the limb as a pair of circumferences, formed by the measurement points of the circumference of the seven points of the arm and forearm, called truncated cones (Sander et al., 2002). 

The method of measuring the indirect volume has acceptable levels of intra- and inter-examiner reliability, with intraclass correlation coefficient (ICC) values of 0.99, and is easily operational (Taylor et al., 2006). The final estimated excess volume corresponds to the sum of the differences between each point. Lymphedema was considered when there was a difference greater than 2 cm in the perimetry of two or more predetermined points on the affected limb compared to the contralateral limb (Sander et al., 2002). This information was inserted in the manuscript and is highlighted in the document 'Revised Manuscript with Track Changes'. 

Sander AP, Hajer NM, Hemenway K, et al (2002) Upper- extremity volume measurements in women with lymphedema: a comparison of measurements obtained via water displacement with geometrically determined volume. Phys Ther. 82:1201–1212

Taylor R, Jayasinghe UW, Koelmeyer L, et al. (2006) Reliability and validity of arm volume measurements for assessment of lymphedema. Phys Ther 86:205–214

2) In the last paragraph of the Discussion, the authors say that the US seems to be a promising tool for the monitoring of interventions, but this study did not assess the changes of US findings after treatment for lymphedema and the prognosis.

Answer: Clinical measures such as volumetry and palpation are still the main measures for monitoring lymphedema treatments. Although volumetry has an excellent ability to measure the difference in volume between the limbs, it is only concerned with measuring an external characteristic. Simultaneously, palpation is not very reliable because it is a dependent evaluator. The subcutaneous tissue so affected by lymphedema, cannot be reliably represented by these clinical measures, so the treatments are focused on just decreasing volume. Although the recommendations of the guidelines for the diagnosis and prognosis of lymphedema may be typically delayed concerning new developments in the diagnostic field, the changes in the guidelines depend on robust evidence, necessary for the diagnostic ultrasound to be an integral part in monitoring the treatment of this dysfunction. Indeed, the immediate availability, the non-invasive nature, and the low cost can make this modality the first choice to establish lymphedema measures in the diagnosis and prognosis. 

3) This study did not include the degree of impairment, therefore the first sentence of the conclusion seems inadequate.

Answer: As suggested in the first sentence of the conclusion, the measurement of the degree of impairment was removed, leaving the identification of biophysical changes. This information was inserted in the manuscript and is highlighted in the document 'Revised Manuscript with Track Changes'. 

Minor points

1) Table 1 law means right?

Answer: In table 1 the word "law" has been replaced by "right". This information was inserted in the manuscript and is highlighted in the document 'Revised Manuscript with Track Changes'.

2) Table 2 -4, abbreviation of MSA and MSNA should be spelled out in the footnote.

Answer: The abbreviations MSA and MSNA were explained in the footnote and is highlighted in the document 'Revised Manuscript with Track Changes'.

6. PLOS authors have the option to publish the peer review history of their article (what does this mean?). If published, this will include your full peer review and any attached files. If you choose "no", your identity will remain anonymous, but your review may still be made public.

Do you want your identity to be public for this peer review? For information about this choice, including consent withdrawal, please see our Privacy Policy.

Reviewer #1: No

Reviewer #2: No

---

## [Editor Report · Decision Letter 1]

7 Feb 2022

RELATIONSHIP BETWEEN LYMPHEDEMA AFTER BREAST CANCER TREATMENT AND BIOPHYSICAL CHARACTERISTICS OF THE AFFECTED TISSUE

PONE-D-20-33153R1

Dear Dr. Perez,

We’re pleased to inform you that your manuscript has been judged scientifically suitable for publication and will be formally accepted for publication once it meets all outstanding technical requirements.

Kind regards,

Tatsuo Shimosawa, M.D., Ph.D.

Academic Editor

PLOS ONE

Additional Editor Comments (optional):

The authors responded and revised the article adequately.
---

## [Editor Report · Acceptance letter]

9 Feb 2022

PONE-D-20-33153R1 

RELATIONSHIP BETWEEN LYMPHEDEMA AFTER BREAST CANCER TREATMENT AND BIOPHYSICAL CHARACTERISTICS OF THE AFFECTED TISSUE 

Dear Dr. Perez:

I'm pleased to inform you that your manuscript has been deemed suitable for publication in PLOS ONE. Congratulations! Your manuscript is now with our production department. 

Kind regards, 

on behalf of

Prof. Tatsuo Shimosawa 

Academic Editor

PLOS ONE